# High-Intensity Interval Training and α-Linolenic Acid Supplementation Improve DHA Conversion and Increase the Abundance of Gut Mucosa-Associated *Oscillospira* Bacteria

**DOI:** 10.3390/nu13030788

**Published:** 2021-02-27

**Authors:** Claire Plissonneau, Frederic Capel, Benoit Chassaing, Marine Dupuit, Florie Maillard, Ivan Wawrzyniak, Lydie Combaret, Frederic Dutheil, Monique Etienne, Guillaume Mairesse, Guillaume Chesneau, Nicolas Barnich, Nathalie Boisseau

**Affiliations:** 1Laboratoire des Adaptations Métaboliques à l’Exercice en conditions Physiologiques et Pathologiques (AME2P), Université Clermont Auvergne, CRNH Auvergne, 63000 Clermont-Ferrand, France; claire.plissonneau@uca.fr (C.P.); marine.dupuit@uca.fr (M.D.); florie.maillard@uca.fr (F.M.); monique.etienne@uca.fr (M.E.); 2M2iSH, UMR 1071 Inserm/Université d’Auvergne; USC-INRAE 2018, Microbes, Intestin, Inflammation et Susceptibilité de l’Hôte (M2iSH), Université Clermont Auvergne, CRNH Auvergne, 63000 Clermont-Ferrand, France; nicolas.barnich@uca.fr; 3INRAE, UNH, Unité de Nutrition Humaine, CRNH Auvergne, Université Clermont Auvergne, 63000 Clermont-Ferrand, France; frederic.capel@inrae.fr (F.C.); lydie.combaret@inrae.fr (L.C.); 4Inserm U1016, Team ‘‘Mucosal Microbiota in Chronic Inflammatory Diseases’’, CNRS UMR 8104, Université de Paris, 75014 Paris, France; benoit.chassaing@inserm.fr; 5LMGE, CNRS, Laboratoire Microorganismes: Génome et Environnement, Université Clermont Auvergne, 63000 Clermont-Ferrand, France; ivan.wawrzyniak@uca.fr; 6CNRS, LaPSCo, Physiological and Psychosocial Stress, University Hospital of Clermont-Ferrand, CHU Clermont-Ferrand, WittyFit, 63000 Clermont-Ferrand, France; frederic.dutheil@uca.fr; 7Valorex, La Messayais, 35210 Combourtillé, France; g.mairesse@valorex.com (G.M.); g.chesneau@valorex.com (G.C.)

**Keywords:** exercise, linseed oil supplementation, body composition, microbiota

## Abstract

Obesity, a major public health problem, is the consequence of an excess of body fat and biological alterations in the adipose tissue. Our aim was to determine whether high-intensity interval training (HIIT) and/or α-linolenic acid supplementation (to equilibrate the n-6/n-3 polyunsaturated fatty acids (PUFA) ratio) might prevent obesity disorders, particularly by modulating the mucosa-associated microbiota. Wistar rats received a low fat diet (LFD; control) or high fat diet (HFD) for 16 weeks to induce obesity. Then, animals in the HFD group were divided in four groups: HFD (control), HFD + linseed oil (LO), HFD + HIIT, HFD + HIIT + LO. In the HIIT groups, rats ran on a treadmill, 4 days.week^−1^. Erythrocyte n-3 PUFA content, body composition, inflammation, and intestinal mucosa-associated microbiota composition were assessed after 12 weeks. LO supplementation enhanced α-linolenic acid (ALA) to docosahexaenoic acid (DHA) conversion in erythrocytes, and HIIT potentiated this conversion. Compared with HFD, HIIT limited weight gain, fat mass accumulation, and adipocyte size, whereas LO reduced systemic inflammation. HIIT had the main effect on gut microbiota β-diversity, but the HIIT + LO association significantly increased *Oscillospira* relative abundance. In our conditions, HIIT had a major effect on body fat mass, whereas HIIT + LO improved ALA conversion to DHA and increased the abundance of *Oscillospira* bacteria in the microbiota.

## 1. Introduction

Obesity is one of the major public health challenges and its prevalence has been increasing worldwide for several decades. In 2016, 39% of adults in the world were overweight and 13% were obese [1]. Obesity is characterized by excessive accumulation of adipose tissue, especially visceral adipose tissue, which is positively correlated with metabolic disorders [2]. Visceral adipose tissue releases pro-inflammatory factors, such as cytokines (TNF-α, IL-6, Il-1β), free fatty acids, and other molecules [3], which lead to chronic low-grade inflammation and promote insulin resistance [4].

Endurance training is an effective strategy to prevent overweight and obesity [5]. This exercise modality includes low- to moderate-intensity continuous training, and high intensity interval training (HIIT). Moderate-intensity continuous training remains the most recommended physical activity [6], but HIIT is now also suggested for people with obesity [7]. HIIT (i.e., repeated bouts of high-intensity effort at >80–85% of the peak heart rate followed by varied recovery times [8]) is a time-efficient and safe strategy to reduce total fat mass (FM), particularly subcutaneous and intra-abdominal FM [9,10]. HIIT programs also decrease inflammation and improve insulin sensitivity [11,12,13]. In addition, several supplements and nutritional interventions may enhance HIIT effects by increasing energy metabolism, or by modulating the adaptive response during recovery [14]. However, no data are available concerning the potential effect of n-3 polyunsaturated fatty acid supplementation on HIIT adaptations.

N-3 polyunsaturated fatty acids (n-3 PUFAs), from natural sources or dietary supplements, also exert a beneficial effect on the body composition and inflammation status [15,16]. Therefore, n-3 PUFA supplementation could be a relevant strategy for managing chronic inflammatory diseases, including obesity-linked low-grade inflammation. Most studies used fish oil to increase eicosapentaenoic acid (EPA; C20:5 n-3) and docosahexaenoic acid (DHA; C22:6 n-3) [17,18,19] rather than the precursor α-linolenic acid (ALA) [20,21], from linseed oil, especially for n-3 PUFA supplementation. EPA and DHA are considered the biologically active components of n-3 PUFAs, and have been associated with weight stabilization [16], lower systemic inflammation [22], and improved lipid profile, especially plasma triglycerides and cholesterol [23,24]. EPA and DHA are found mainly in seafood and can be generated, to a limited extent, by the post-ingestion conversion of α-linolenic acid (ALA, C18:3 n-3), which is mainly found in plant-based products.

The intestinal microbiota could be another potential therapeutic target because gut microbiota dysbiosis has been associated with adiposity excess and visceral inflammation [25,26]. Gut microbiota composition and function could be influenced by regular physical activity, and also by n-3 PUFAs intake [17,27]. It has been shown that exercise restores gut microbiota equilibrium by increasing the ratio of beneficial bacteria relative to pathogenic bacteria and their diversity [27,28,29]. Regular physical activity also protects against intestinal barrier and permeability dysfunctions [30]. Similarly, n-3 PUFAs improve intestinal barrier function and integrity [31] and increase healthy bacterial communities [17,18,32].

On the basis of these data, the aim of this study was to evaluate the effects of a 12-week intervention program that included physical activity (HIIT), n-3 PUFA supplementation through addition of linseed oil (LO) in the diet, or both (HIIT + LO) on body composition and metabolic profile changes in a rodent model of obesity. We first hypothesized that HIIT and LO positively affect whole body FM, mesenteric (visceral) FM, and systemic inflammation, and that their combination would induce greater positive effects. We also hypothesized that each intervention could specifically affect the mucosa-associated gut microbiota composition (α and β diversity) with more favorable adaptations in the HIIT + LO group. A better understanding of the effects of physical activity, n-3 PUFA consumption, and their potential interaction will help to develop personalized intervention strategies to reduce adiposity, inflammation, and gut microbiota dysbiosis.

## 2. Materials and Methods

### 2.1. Animals

Sixty 8-week-old male Wistar rats (Charles River Laboratory, France) were used for this study. They were individually housed in a temperature-controlled (22 ± 2 °C) room with reversed 12 h light/dark cycle and free access to food and water. All experiments were approved by the local ethics committee (C2EA-02, Auvergne, France; APAFIS 16090-2018071208306750), and were performed in accordance with the European animal welfare regulations and guidelines (European Directive 2010/63/EU on the protection of vertebrate animals used for experimental and scientific purposes). All efforts were made to protect animal welfare and to minimize suffering at each step of the protocol. Animals were sacrificed by cervical dislocation after isoflurane anesthesia.

### 2.2. Experimental Design

Phase 1: obesity induction (16 weeks). Rats were randomly assigned to one of the two groups: low-fat diet (LFD; 11% of total kcal as fat) (*n* = 12) and high-fat diet (HFD; 43% of total kcal as fat to induce obesity) (*n* = 48) (Figure 1). In both groups, food was consumed ad libitum for 16 weeks.

Phase 2: training and/or diet intervention (12 weeks). After these first 16 weeks, the HFD group was divided in four groups matched for body weight and total fat mass: (i) HIIT group (*n* = 12) that performed the exercise program; (ii) LO group (*n* = 12) that received LO supplementation; (iii) HIIT + LO group (*n* = 12); and (iv) HFD group (*n* = 12) without LO and HIIT (control). The four groups continued to receive the same HFD as during phase 1 (Figure 1).

### 2.3. Diet Composition

The diets were prepared by INRAE (Jouy-en-Josas, France). The LFD included proteins (19.8% of all kcal), carbohydrates (68.7%), and fat (11.5%). The HFD was composed of proteins (17.3% of all kcal), carbohydrates (39.4%), and fat (including 4% of sunflower oil; 43.3% of all kcal). The diet of the LO group had the same composition as the HFD, but 2% of sunflower oil was replaced by 2% of LO. All diets were analyzed by gas chromatography to confirm their n-6/n-3 PUFAs ratio (Table 1).

### 2.4. Training Design

Rats in the HIIT (*n* = 12) and HIIT+LO (*n* = 12) groups underwent a 2-week acclimation on a treadmill (Matsport, France) with a progressive increase of speed and running time. “No runner” rats were excluded at the end of this period. Then, rats performed a 12-week HIIT program, 4 days per week, as follows: 6 × (3 min at 10 m.min^−1^ × 4 min at 18 m.min^−1^), for a total of 42 min/training (Figure 1).

### 2.5. Body Composition

Body composition (weight, FM, and fat free mass (FFM)) were measured using an EchoMRI device (EchoMRI Medical System, Houston, TX, USA) at week 1 and 16 during phase 1 (obesity induction), and at week 22, 25, and 28 during phase 2 (training and/or LO supplementation). Body weight was measured three times per week using a standard scale. Area of adipocytes (µm) was measured with the ImageJ software (Rasband, W.S., ImageJ, U.S. National Institutes of Health, Bethesda, MD, USA).

### 2.6. Oral Glucose Test Tolerance

Glucose sensitivity was measured with an oral glucose tolerance test (OGTT) at the end of the induction phase (week 16) and at the study end (week 28) after overnight fasting. Glucose level was measured in blood samples collected from the tail vein using a standard glucometer (Xpress, Nova Biomedical, Waltham, MA, USA) at baseline, and then at 15, 30, 60, 90, and 120 min after glucose administration by gavage (2 g.kg^−1^ FFM). The area under the curve (AUC) for glucose and the netAUC (after subtraction of the baseline glucose concentration) were calculated.

### 2.7. Plasma and Erythrocytes

Blood samples were collected by aorta puncture. After centrifugation, plasma was collected in EDTA tubes and stored at −80 °C before biochemical analyses. Erythrocytes were purified from blood samples collected in tubes saturated with EDTA (EDTA K3 tubes). Samples were centrifuged (3500 rpm, 4 °C, 10 min) and the supernatant (plasma) was discarded and replaced by the equivalent amount of cold acidified saline solution (100 µL H_2_NSO_4_/100 mL). Erythrocytes were resuspended, centrifuged, and supernatant was removed. This step was repeated twice. Washed erythrocytes were then stored at −80 °C before analysis.

### 2.8. Fatty Acid Profiling in Erythrocytes and Diets

Fatty acid profiling was performed by gas chromatography–flame ionization detection (GC–FID). Briefly, total lipids were extracted from food or erythrocytes using the Folch or method [33,34]. The organic phase was evaporated under nitrogen for fatty acid methylation. Fatty acid methyl ester (FAME) separation was done by gas chromatography (GC) with a select FAME (Agilent technologies, Les Ulis, France) column (0.25 mm* i.d., 100 m., 0.25 μm film thickness) on a GC system (Thermo Trace Finnigan GC ultra; Waltham, MA, USA) equipped with a flame ionization detector. FAMEs were identified using different commercial FAME standards from Supelco (Sigma) and Nu-Chek Prep (Elysian, MN, USA). Peak integration was performed with Chromeleon (Version 7.2.4, Dionex, Thermo Scientific, Courtaboeuf, France).

### 2.9. Measurement of Plasma Myeloperoxidase and Lipopolysaccharide Levels

Plasma myeloperoxidase (MPO) concentrations were measured using a commercial ELISA kit (Myeloperoxidase DuoSet ELISA, R&D Systems, Minneapolis, MN, USA). Triglycerides were measured in hepatic samples using a commercial kit (Cayman Chemical, Ann Arbor, MI, USA). Plasma lipopolysaccharide (LPS) activity was evaluated in HEK-Blue-mTLR4 cells (Invivogen, San Diego, CA, USA). Briefly, 180 µL of cell suspension (1.4 × 10^4^ cells per mL of HEK-Blue Detection medium) (Invivogen, San Diego, CA, USA) was added to 20 µL of each diluted (1:1000) plasma sample. LPS (Invivogen, San Diego, CA, USA) was used as positive control and standard range. Plates were incubated at 37 °C in 5% CO_2_ for 24 h, and alkaline phosphatase activity was measured at 620 nm.

### 2.10. Protein Extraction and Western Blotting

Mesenteric and colon samples were homogenized in 1 mL lysis buffer (25 mM Tris, 5 mM EDTA, 0.1 mM MgCl_2_, 10% glycerol, 150 mM NaCl, 1% Nonidet P-40, 1% SDS) with freshly added protease inhibitor cocktail (Complete, Mini, EDTA-free Protease Inhibitor Cocktail, Roche), 1 mM sodium orthovanadate, 1 mM PMSF, and 5 mM N-ethylmaleimide. Homogenates were centrifuged at 10,000 rpm at 4 °C for 5 min. An aliquot (25 µL) was used for protein concentration quantification with the optical density protein assay (Bio-Rad, Hercules, CA, USA). Proteins were stored at −80 °C until analysis.

Protein were separated on 4–15% gels (Mini-PROTEAN TGX Stain-Free Protein Gels, Bio-Rad, Hercules, CA, USA), then transferred to PVDF membranes (Bio-Rad). Membranes were blocked with 5% BSA in TBST (pH 8 Tris buffered saline/0.05% Tween-20) at room temperature under agitation for 1 h. Membranes were incubated with diluted primary antibodies against zonula occludens-1 (ZO-1) (Thermo Fisher, polyclonal, 1:500 dilution) and occludin (Thermo Fisher, monoclonal, 1:500 dilution), at 4 °C under agitation, overnight. After three washes in TBST, membranes were incubated with secondary antibodies at room temperature for 1h. Antibody interactions were detected with the Enhanced Chemiluminescence Detection Kit (Clarity Western ECL Substrate, Bio-Rad, Hercules, CA, USA) and the Bio-Rad ChemiDoc imaging system. Data were normalized to the total protein amount using the Stain-Free blot system (Bio-Rad system). Band intensity was analyzed with Image Lab (Bio-Rad).

### 2.11. Fecal Short-Chain Fatty Acid Quantification

Weighted rat fecal samples were reconstituted in 200 µL MilliQ^®^ water. Samples were homogenized, incubated at 4 °C for 2 h, and then centrifuged at 12,000× *g* at 4 °C for 15 min. Supernatants were weighted and saturated phosphotungstic acid (100 µL for 1 g of fecal content) was added to the samples. Samples were incubated at 4 °C overnight, centrifuged, and then short-chain fatty acid (SCFA) concentrations (including butyrate, propionate, and acetate) were determined by gas chromatography (Nelson 1020, Perkin-Elmer, St Quentin en Yvelines, France). Chromatographic separation was achieved on DB-FFAP columns (30 m × 250 μm, 0.25 μm). The inlet temperature was 250 °C and the injection volume was 1 μL. The initial oven temperature was 100 °C, and then was ramped to 250 °C (10 °C/min) and held for 5 min. The carrier gas was helium at a constant flow of 7 mL/min. Samples were imported to the column using the split mode at a ratio of 10:1. Detection was performed with the FID method.

### 2.12. Microbiota Composition Analyses

For genomic DNA extraction, rat colon samples were lysed in proteinase K at 56 °C in a shaking incubator overnight. DNA was extracted using the NucleoSpin^®^ Tissue kit (Macherey-Nagel, Germany). DNA concentration was determined with a Qubit^TM^ fluorometer (Invitrogen), and the DNA quality was evaluated with a NanoDrop™ (Thermo Scientific) spectrophotometer (260/280 and 260/230 ratio). Region V4 of the 16S rRNA gene was PCR-amplified from each sample using forward and reverse primers that were designed with the Golay error-correcting scheme and used to tag PCR products from individual samples [35]. The sequence of the composite forward primer 515F was: 5′- *AATGATACGGCGACCACCGAGATCTACACGCT*XXXXXXXXXXXXTATGGTAATT*GT*GTGYCAGCMGCCGCGGTAA-3′. The italicized sequence represents the 5′ Illumina adapter, the sequence of 12 Xs is the Golay barcode, the bold sequence is the primer pad, the italicized and bold nucleotide is the primer linker, and the underlined sequence is the conserved bacterial sequence. The sequence of the reverse primer 806R was 5′-*CAAGCAGAAGACGGCATACGAGAT*AGTCAGCCAG*CC*
GGACTACNVGGGTWTCTAAT-3′. The italicized sequence is the 3′ reverse complement sequence of the Illumina adapter, the bold sequence is the primer pad, the italicized and bold nucleotides are the primer linker, and the underlined sequence is the conserved bacterial sequence. PCR mixtures included the Hot Master PCR mix (Quantabio, Beverly, MA, USA), 0.2 µM of each primer, and 10–100 ng of template. The reaction conditions were 3 min at 95 °C, followed by 30 cycles of 45 s at 95 °C, 60 s at 50 °C, and 90 s at 72 °C on a Bio-Rad thermocycler. The PCR products were purified with Ampure magnetic purification beads (Agencourt, Brea, CA, USA) and visualized by gel electrophoresis. The products were then quantified (BIOTEK fluorescence spectrophotometer) using the Quant-iT PicoGreen dsDNA assay. A master DNA pool was generated from the purified products mixed in equimolar ratios. The pooled products were quantified using the Quant-iT PicoGreen dsDNA assay and then sequenced using an Illumina MiSeq sequencer (paired-end reads, 2 × 250 bp) at Cornell University, Ithaca.

Forward and reverse Illumina reads were joined using the fastq-join method, and sequences were demultiplexed and their quality filtered using the Quantitative Insights Into Microbial Ecology 2 (QIIME, version 2, USA) software [36]. QIIME default parameters were used for quality filtering: reads truncated at the first low-quality base were excluded if (1) there were more than three consecutive low-quality base calls; (2) less than 75% of the read length was made of consecutive high-quality base calls; (3) at least one uncalled base was present; (4) more than 1.5 errors were present in the barcode; (5) any Phred quality was below 20; or (6) the length was less than 75 bases. Sequences were assigned to operational taxonomic units (OTUs) using the UCLUST algorithm [37] with a 97% threshold for pairwise identity (with the creation of new clusters with sequences that did not match the reference sequences) and classified taxonomically using the Greengenes reference database 13_8 [38]. A single representative sequence for each OTU was aligned, and a phylogenetic tree was built using FastTree [39]. The phylogenetic tree was used to compute the unweighted UniFrac distances between samples [40,41], and the rarefaction method to compare OTU abundances across samples. Principal coordinates analysis (PCoA) plots were used to assess the variation among groups (β diversity).

### 2.13. Statistical Analysis

All statistical analyses were performed with the Statistica software (version 12). Data were presented as the mean ± standard deviation (SD). Normal data distribution was tested using the Kolmogorov–Smirnov test and the homogeneity of variance with the F-test. In the absence of normal distribution or variance homoscedasticity, data were log-transformed before analysis. Phase 1 (obesity induction): one-way ANOVA with the two experimental groups (LFD and HFD) as the main factor was performed using the Newman–Keuls post-hoc test to determine differences between groups. Phase 2 (training and/or LO supplementation): one-way ANOVA (with or without repeated measures) with the four experimental groups (HIIT, HIIT + LO, LO, HFD) was performed using the Newman–Keuls post-hoc test to determine group effects or time (T), group (G), and T*G interactions. Moreover, a 2-way ANOVA was used to determine the main effect of exercise (HIIT), LO supplementation, and their interaction, on the HFD [42]. Spearman correlations were used to test relationships between variables. Differences were considered significant when *p*-values < 0.05.

## 3. Results

### 3.1. The HFD Induces a Pre-Obesity State in Wistar Rats

Rats were randomly separated in two groups at arrival: LFD (*n* = 12; controls) and HFD (*n* = 48; 16-week obesity induction) (Figure 1). The cumulative food intake (kcal) during this period did not significantly differ between groups (data not shown). Weight was monitored during the 16 weeks of phase 1 and became significantly higher in the HFD group from week 14 (Figure 2A). At the end of phase 1 (week 16), total FM, but not FFM, was significantly higher in the HFD than LFD group (75.0 ± 20.6 and 56.6 ± 14.0 g; *p* < 0.05) (Figure 2B). Although fasting glycemia was not different between groups, blood glucose values during the OGTT were higher in the HFD than LFD group (Figure 2C). The netAUC value also was significantly higher in the HFD than LFD group (*p* < 0.05) (Figure 2D). Thus, 16 weeks of HFD in 8-week-old Wistar rats induced higher total FM gain and decreased glucose tolerance, leading to a pre-obesity status.

After the 16-week induction period, animals in the HFD group were divided in four homogeneous groups (*n* = 12) matched for weight and FM (Figure 1). The 12 rats in the HFD group became the control group to assess the effect of training and/or LO supplementation during the next 12 weeks.

### 3.2. LO Supplementation Enhances DHA Conversion in Erythrocytes That Is Potentiated by HIIT

In the LO and HIIT+LO groups, the HFD was supplemented with LO (isocaloric substitution of sunflower oil with LO) to increase the intake of n-3 PUFAs and decrease that of n-6 PUFAs without any change in the total amount of ingested lipids. Quantification of n-3 PUFAs in erythrocyte membranes showed that their percentage (relative to the total fatty acid amount) was higher in erythrocytes from rats in the LO and HIIT+LO groups compared with the HFD and HIIT groups (without LO supplementation) (Figure 3A). As LO is rich in ALA, but does not contain EPA and DHA, our results showed that ALA was converted to EPA and DHA in membrane phospholipids of mice supplemented with LO, as indicated by the higher EPA and DHA percentages in these two groups compared with the HFD and HIIT groups (*p* < 0.05) (Figure 3A). Moreover, DHA concentration was higher in the HIIT + LO than LO group (LO: 1.69 ± 0.28% and HIIT + LO: 2.16 ± 0.46%; *p* < 0.01) (Figure 3C). Two-way ANOVA showed a significant interaction between HIIT and LO supplementation (*p* < 0.05), with 2.7-fold lower n-6/n-3 PUFA ratio values in the HIIT + LO than LO group (Figure 3D). The level of linoleic acid (LA), n-6 PUFA precursor, did not differ among groups (*p* = 0.17). Conversely, arachidonic acid (AA, derived from LA) content was lower in the LO than in the HFD and HIIT groups (Figure 3B). This change in PUFA composition had no impact on the cumulative food intake (kcal) during phase 2 (data not shown).

Thus, LO supplementation increased EPA and DHA content in erythrocyte membranes. Interestingly, HIIT enhanced LO conversion especially to DHA, and therefore could potentiate the beneficial effects of LO supplementation.

### 3.3. HIIT Changes the Body Composition and Adipocyte Cell Size

Only the HIIT program, but not LO supplementation, changed body composition. At the end of phase 2, body weight gain was much smaller in the HIIT and HIIT + LO groups than in the HFD and LO groups (*p* < 0.001) (Figure 4A). This was explained by the lower FM gain in these two actively exercising groups (Figure 4A). FFM did not increase in the HIIT and HIIT + LO groups compared with the HFD and LO groups (Figure 4A). On the other hand, FM gain in the mesenteric adipose tissue was not different among groups (Figure 4B). LO supplementation alone had no effect on weight, total FM, and mesenteric FM, but significantly decreased the mesenteric adipocyte size compared with the HFD group (*p* < 0.05) (Figure 4D). Training (i.e., HIIT and HIIT + LO groups) also decreased adipocyte size in the mesenteric and subcutaneous adipose tissues (Figure 4C). The HIIT + LO combination did not have any additional effect on adipocyte size.

Metabolically, hepatic triglycerides were significantly decreased only in the HITT group compared with the LO group (*p* < 0.05) (data not shown). Fasting glycemia was significantly decreased in both HIIT groups (*p* < 0.05), but not in the LO group. Glucose tolerance was similar among groups at the study end (Figure 5). Thus, the HIIT program (regular physical exercise) had a significant effect on FM gain, adipocyte size, and fasting glycemia, whereas LO supplementation did not have such effects and did not potentiate HIIT effects.

### 3.4. LO Supplementation Limits Systemic Inflammation

As a HFD can alter the barrier function leading to LPS release and endotoxemia, the effect of LO supplementation, combined or not with HIIT, on tight junction protein expression was then investigated. At the end of phase 2, the expression of ZO-1 and occludin was not different in the four groups (Table 2). However, LPS concentration in plasma, which could reflect intestinal permeability, was not different between groups (Figure 6). Nevertheless, LPS was negatively correlated with the tight junction protein ZO-1. Plasma MPO concentration, a marker of systemic inflammation, also was significantly decreased in the LO and HIIT/HIIT + LO groups compared with the HFD group (*p* < 0.05). However, the “HIIT+LO” combination did not further reduce the systemic inflammation. Together, these data suggest that n-3 PUFAs could limit systemic inflammation in Wistar rats with HFD-induced pre-obesity.

### 3.5. HIIT Combined with LO Modulates the Intestinal Mucosa-Associated Microbiota

HFD consumption induces profound alterations of the intestinal microbiota composition. At the end of phase 2, analysis of the colon mucosa microbiota composition by 16S rRNA sequencing showed that the α-diversity (Shannon diversity index) was comparable in the four groups of pre-obese rats (Figure 7A). However, the β-diversity, computed by PCoA on unweighted UniFrac distance matrices, indicated that rat colon samples clustered in two main groups: HIIT (HIIT and HIIT + LO) and no HIIT (HFD and LO) (Figure 7C). HIIT had the main effect on the composition of the mucosa-associated microbiota, whereas LO on its own did not have any effect on microbiota β-diversity. On the other hand, the HIIT + LO combination induced a significant β-diversity difference when compared with the HIIT group (*p* < 0.05).

The HIIT and LO combination may differently influence metabolites or specific features compared with HIIT and LO on their own. As SCFAs are derived from the microbiota, major SCFAs (acetate, butyrate, and propionate) were quantified in fecal samples to evaluate the microbiota global fermentative-metabolic capacity. Acetate, butyrate, and propionate concentrations were not different among groups after the 12-week intervention as well as the total SCFA amount (acetate + butyrate + propionate) (Table 3).

Concerning the specific abundance of microbiota phyla, the *Tenericutes* phylum was more abundant in the HIIT groups (*p* < 0.05), and Proteobacteria showed a similar trend (*p* = 0.06). *Verrucomicrobia* tended to decrease (*p* = 0.09), especially in the HIIT + LO group (Figure 7D). No other global phylum variation was found. However, the ANCOM statistical analysis highlighted significant group differences (Figure 8). The relative abundance of *Clostridiales* spp. was lower in the HIIT and HIIT+LO than in the HFD and LO groups (*p* < 0.05) (Figure 8A). Conversely, the relative abundance of *Prevotella* (linked to diets rich in fibers and physical activity), *Anaeroplasma* and Cyanobacteria YS2 was significantly higher in the HIIT and HIIT + LO than in the HFD and LO groups (*p* < 0.05), and was mainly related to HIIT (Figure 8B–D). Finally, *Oscillospira* relative abundance was significantly higher in the HIIT + LO group than in the other groups (*p* < 0.05) (Figure 8E). The relative abundance of specific microbiota components was correlated with body composition (FM and FFM), but not with inflammation changes (MPO level). Weight, FM, and FFM gains were positively correlated with *Clostridiales* spp. relative abundance, and negatively correlated with *Anaeroplasma*, Cyanobacteria YS2, *Prevotella,* and *Oscillospira* relative abundance (Figure 9).

## 4. Discussion

The aim of this study was to determine whether HIIT, combined or not with LO supplementation, alters the body composition and metabolic profile in a rat model of pre-obesity, specifically by modulating the intestinal mucosa-associated microbiota. In our experimental conditions, we found that HIIT significantly reduced the body FM, and that the HIIT + LO combination improved ALA to DHA conversion and increased the relative abundance of *Oscillospira* bacteria in the colon microbiota.

Obesity is the consequence of an excess of body fat and of biological alterations in the adipose tissue, leading to health disorders, and potentially reducing life expectancy [2,43]. Its causes are complex and results from the interweaving of several (dietary, genetic, epigenetic, and environmental) factors implicated in the development and progression of this chronic disease [44,45,46,47]. Dietary changes and physical inactivity play an indisputable role in the recent emergence of obesity. Therefore, we chose a model of obesity induced by a Western-style diet, rather than a genetic model, to better mimic the current situation. The 16-week HFD led to a pre-obesity phenotype in 8-week-old Wistar rats, as previously shown by Gerbaix et al. [48]. Weight and FM gains were higher in the HFD group than in the LFD group, and glucose tolerance was altered only in the HFD group. In humans, excessive lipid intake is one of the factors that favor weight and FM gain [44,47]. In western countries, although the total amount of lipid consumption has remained quite constant in the last twenty years, their quality has changed, particularly the n-6/n-3 PUFA ratio [49]. When n-3 PUFAs are provided in insufficient proportion relative to n-6 PUFAs, n-6 PUFA derivatives (i.e., prostaglandins) are in excess, stimulate the multiplication of adipocytes, and have a long-term pro-obesogenic effect [50]. Conversely, n-3 PUFAs facilitate fat burning, especially in the visceral area [51]. Therefore, in the phase 2 of this study, we rebalanced the n-6/n-3 PUFA ratio in the HFD, by providing n-3 PUFAs of plant origin (linseed) and not of animal origin (fish), and assessed the impact of such change on obesity prevention/treatment. LO supplementation increased the total n-3 PUFA percentage in erythrocytes, which is a reliable long-term biomarker of the diet fatty acid profile and incorporation in tissues [52]. The incorporation of ALA, EPA, and DHA was increased in the LO and HIIT + LO groups, but DHA conversion was enhanced when LO supplementation was associated with exercise. Due to the low ALA conversion to EPA and DHA, most studies used fish oil to increase EPA and DHA percentage [17,18,19] rather than plant-derived oils [20,21]. Moreover, in most studies, supplementation was at supra-physiological doses to potentiate PUFA effect. In our study, the aim was to restore a healthy n-6/n-3 PUFA ratio [53,54]. Even at this physiological intake, total n-3 PUFAs were significantly incorporated in erythrocyte cell membranes, which validated our model. However, more studies are needed to understand the mechanisms underlying n-3 PUFA conversion and the impact of physical activity.

A growing body of evidences shows that HIIT is a time-efficient exercise mode for loosing total and visceral FM in humans and rodents [7,9,55,56]. Kapravelou et al. demonstrated that a HIIT protocol (4 min at 65–80% of VO_2_ max followed by 3 min at 50–65% of VO_2_ max for 8 weeks) decreases total and abdominal FM in old Zucker rats [56]. In a recent study performed using the same exercise protocol, our group demonstrated that after 10 weeks of exercise, total and epididymal FM were significantly reduced in obese Zucker rats that performed a HIIT program, but not a classical moderate intensity continuous training (MICT) program [11]. With the current model of obesity that better mimic real environmental conditions, we confirmed that a 12-week HIIT program limits weight and FM gain and decreases adipocyte size in the mesenteric and subcutaneous adipose tissues.

N-3 PUFA supplementation also may affect lipid metabolism, potentially promoting body composition changes [24]. Martins et al. showed that oral gavage with fish oil (three times per week at 2 g per kg body weight) decreases weight and FM gains in mice fed an obesogenic diet [57]. N-3 PUFA supplementation may modulate fatty acid oxidation and/or the secretion of several adipokines, such as leptin [24]. Leptin activates hypothalamic-mediated appetite suppression in response to caloric intake, but its activity is not restricted to this anorexigenic effect. Fatty acids might regulate the expression of the leptin gene and of other adipocyte-specific genes by modulating the regulatory activity of the peroxisome proliferator-activated receptor γ (PPARγ) [58]. A decrease of PPARγ activity stimulates body weight and FM loss and reduces food consumption [59]. Here, we did not observe any LO impact on body composition, despite the effective integration of n-3 PUFAs in erythrocyte cell membranes. The choice of supplementing the diet with a physiologic dose of n-3 PUFAs may partly explain these results because the 12-week LO intervention did not show any anorexigenic effect and did not increase plasma leptin levels (data not shown). LO supplementation did not potentiate HIIT-induced body composition changes. Thus, we can speculate that our training protocol prevents FM gain more efficiently than LO supplementation.

MPO, one of the active substances released by polymorph nuclear leucocytes, is considered a good indicator of systemic inflammation [60]. Interestingly, plasma MPO level was decreased in the LO supplemented groups (LO and HIIT + LO). N-3 PUFAs may reduce leucocyte adhesion and infiltration, and, consequently, systemic inflammation. These results are consistent with previous studies showing that plasma MPO level decreases after n-3 PUFA supplementation in patients with chronic kidney disease, and this might limit chronic inflammation [61]. Moreover, an olive oil-based diet with fish oil supplementation for 2 weeks before colitis induction with dextran sodium sulfate decreased MPO activity in colon [62]. On the other hand, we found that HIIT had no effect on plasma MPO, and its association with LO did not enhance LO effect on MPO concentration in the HIIT + LO group.

As obesity is associated with changes in the microbiota composition and structure, we then evaluated the impact of HIIT and/or LO supplementation on the microbiota associated with the intestinal mucosa. In our study, HIIT + LO supplementation modulated β-diversity of the mucosa-associated microbiota. Our analysis suggests that HIIT had the main effect on microbiota β-diversity, and that LO influences β-diversity only when associated with HIIT. On the other hand, α-diversity was comparable in the four groups. This means that HIIT changed the mucosa-associated microbiota composition without modifying its richness and evenness. The LO amount and/or the supplementation duration might have been not sufficient to alter the global composition of the gut microbiota. Moreover, our supplementation with ALA makes difficult the comparison with previous studies that mainly used EPA and DHA supplementation [17,18,19,63]. Moreover, ground or extruded linseed studied as a matrix could affect differently the mucosa-associated microbiota compared with ALA intake through LO.

The analysis of specific features of the mucosa-associated microbiota highlighted several differences among the four groups. First, *Clostridiales* spp. abundance was significantly decreased only in the HIIT and HIIT + LO groups (but not in the LO group), and this was correlated with weight and FM gain and hepatic triglycerides. Turnbaugh et al., [64] found an increase in *Firmicutes* and *Clostridiales* spp. in individuals with obesity, and Ghosh et al. [17] reported that in mice fed an EPA + DHA-enriched HFD, a physiological intake of EPA and DHA decreases the *Clostridiales* spp. abundance. *Anaeroplasma* and *Cyanobacteria* YS2 were increased in the HIIT and HIIT + LO groups. There is no information on their abundance associated with physical activity and n-3 PUFAs. According to Beller et al. [65], *Anaeroplasma* could be a new potential anti-inflammatory probiotic for the treatment of chronic intestinal inflammation. *Prevotella* also was increased in the HIIT and HIIT + LO groups compared with the HFD and LO groups, and its abundance was negatively correlated with weight and FM gain. Petersen et al. [66] found that *Prevotella* abundance is increased in microbiota of competitive cyclists and this was correlated with the reported exercise time during an average week. In the literature, *Prevotella* has been associated with fiber-rich diets and carbohydrates [67,68] and its abundance may be lower in people following a Mediterranean diet, rich in n-3 PUFAs [69]. It could be interesting to assess the effect a HIIT program associated with a ground linseed-enriched diet (i.e., rich in fibers) on *Prevotella* abundance. Finally, *Oscillospira* abundance was significantly increased only in the HIIT + LO group. The interaction between physical activity and LO supplementation had a greater impact than HIIT or LO alone compared with HFD. Interestingly, *Oscillospira* abundance was negatively correlated with weight and FM gain, but not with FFM gain. In line with our results, different studies showed a negative correlation between *Oscillospira* and body mass index [70,71] and a positive correlation with leanness [72]. Moreover, Maillard et al. found that *Oscillospira* abundance is increased in microbiota of mice after spontaneous physical activity [73]. Finally, Petriz et al. [74], supported by Allen et al. [75], reported a correlation among *Oscillospira* abundance, lactate levels, and exercise intensity. Interestingly, Haro et al. [69] showed higher *Oscillospira* abundance in people with obesity following a Mediterranean diet. Thus, *Oscillospira* strains could be promising new generation probiotics in the context of obesity.

## 5. Conclusions

Obesity is a major public health problem worldwide. Identifying non-drug strategies to prevent and treat this pathology is essential. Our preclinical study using an animal model of obesity induced by a Western diet showed that physical activity plays a major role in limiting weight and FM gain, whereas a diet with a balanced n-6/n-3 PUFA ratio reduces systemic inflammation. HIIT and LO alone or in combination did not lead to major changes of the microbiota associated with the intestinal mucosa, but some bacterial types, particularly Oscillospira, were increased when LO and HIIT were combined. Our study also showed that HIIT associated with LO supplementation enhanced DHA conversion helping to achieve the recommended nutritional intakes. Thus, the combination of HIIT and α-linolenic acid seems favorable and could be proposed in the management of metabolic diseases, such as obesity.

## Figures and Tables

**Figure 1 nutrients-13-00788-f001:**
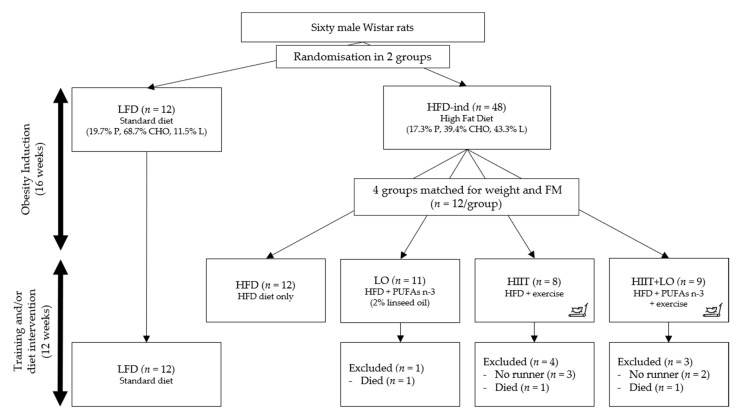
Study design. Rats were randomized in two groups: LFD (*n* = 12) and HFD (*n* = 48) for 16 weeks (phase 1). After this period, the HFD group was divided in four groups (*n* = 12/group) matched for weight and fat mass: HFD, LO, HIIT, and HIIT + LO for 12 weeks (phase 2). In the HIIT and HIIT + LO groups, “no runner” rats were excluded from the study. LFD: low-fat diet, HFD-ind: high-fat diet induction, HFD: high-fat diet, LO: linseed oil, HIIT: high-intensity interval training, FM: fat mass, PUFAs: polyunsaturated fatty acids, P: proteins, CHO: carbohydrates, L: lipids.

**Figure 2 nutrients-13-00788-f002:**
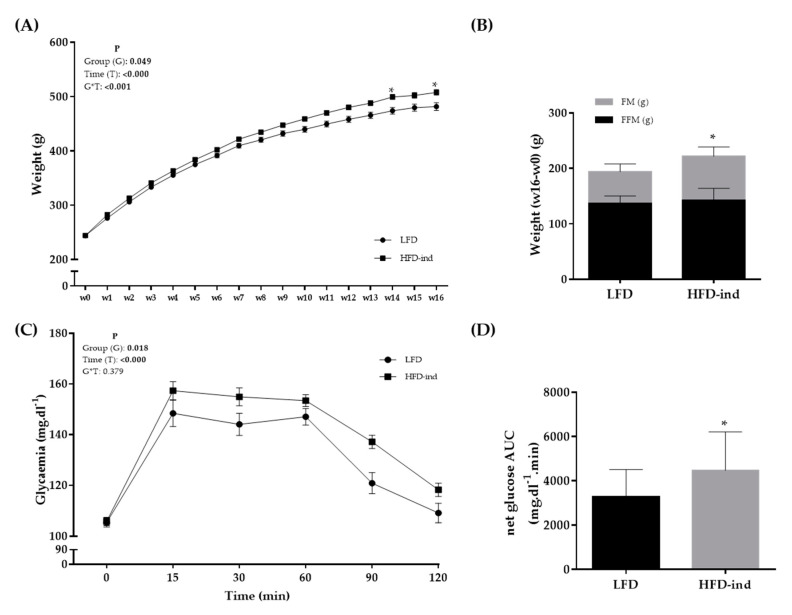
Body composition changes and glycemic profile of the LFD (*n* = 12) and HFD (*n* = 48) groups at the end of phase 1 (obesity induction with the HFD for 16 weeks). (**A**) Weight (g) monitoring during phase 1. (**B**) Weight, fat mass, and fat-free mass changes (w16–w0) in the LFD and HFD groups. (**C**) Glycemia monitoring during the oral glucose tolerance test (mg.dL^−1^). (**D**) NetAUC for glucose (mg.dL^−1^ min) in the LFD and HFD groups. * *p* < 0.05. LFD: low-fat diet, HFD: high-fat diet, AUC: area under the curve, FM: fat mass, FFM: fat free mass.

**Figure 3 nutrients-13-00788-f003:**
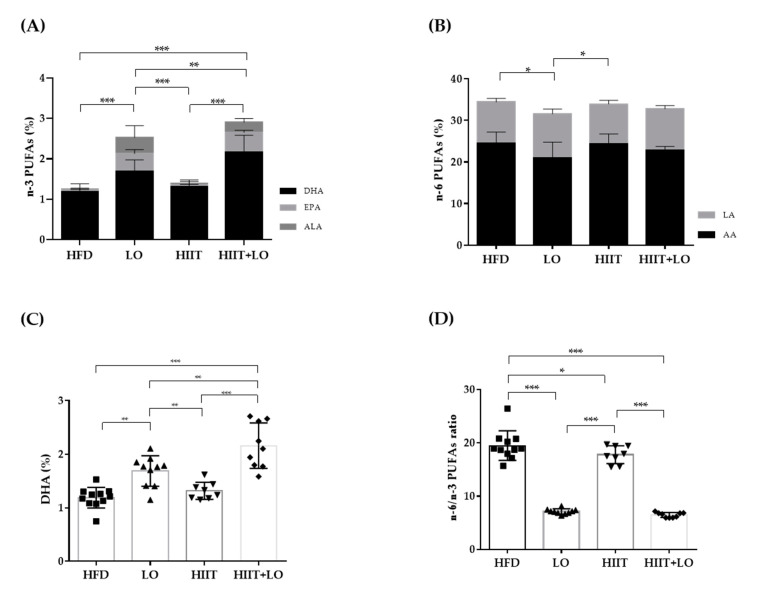
Fatty acid composition in erythrocytes from the four groups: HFD (*n* = 12), LO (*n* = 11), HIIT (*n* = 8), and HIIT+LO (*n* = 9). (**A**) n-3 PUFAs (% of total fatty acids), and ALA, EPA, and DHA distribution (%) in erythrocytes. (**B**) n-6 PUFAs (% of total fatty acids), and LA and AA distribution (%) in erythrocytes. (**C**) DHA (%) in erythrocytes. (**D**) n-6/n-3 PUFAs ratio in erythrocytes. * *p* < 0.05, ** *p* < 0.01, *** *p* < 0.001; PUFAs: polyunsaturated fatty acids, LA: linoleic acid (C18:2 n-6), AA: arachidonic acid (C22:4 n-6), ALA: α-linolenic acid (C18:3 n-3), EPA: eicosapentaenoic acid (C20:5 n-3), DHA: docosahexaenoic acid (C22:6 n-3).

**Figure 4 nutrients-13-00788-f004:**
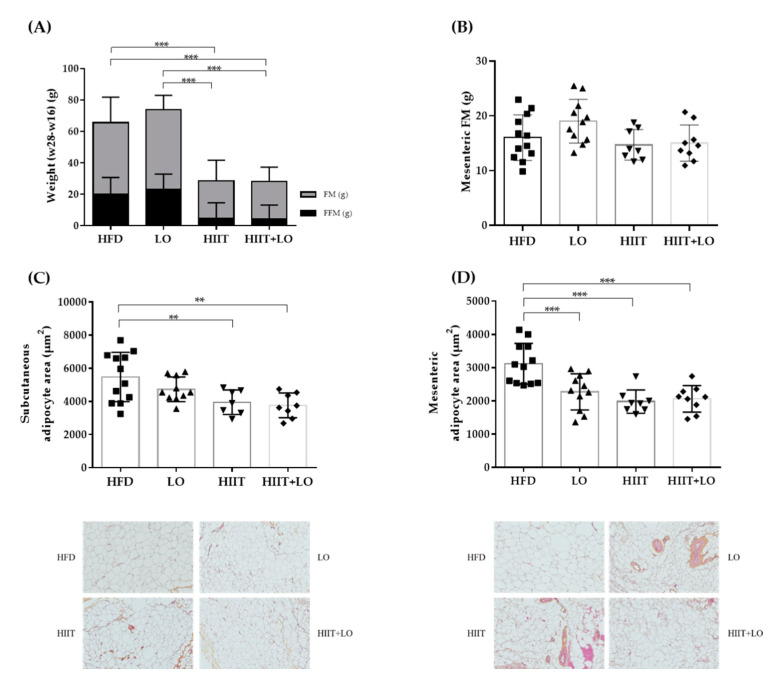
Body composition and adipocyte profiles in the four rat groups: HFD (*n* = 12), LO (*n* = 11), HIIT (*n* = 8), and HIIT + LO (*n* = 9) at the end of phase 2 (training and/or LO supplementation for 12 weeks). (**A**) Weight, fat mass (FM) and fat-free mass (FFM) changes (week 28–week 16) in the four groups (**B**) mesenteric fat mass (FM; g) in the four groups. (**C**) Subcutaneous and (**D**) mesenteric adipocyte area (µm^2^) in the four groups. ** *p* < 0.01 and *** *p* < 0.001.

**Figure 5 nutrients-13-00788-f005:**
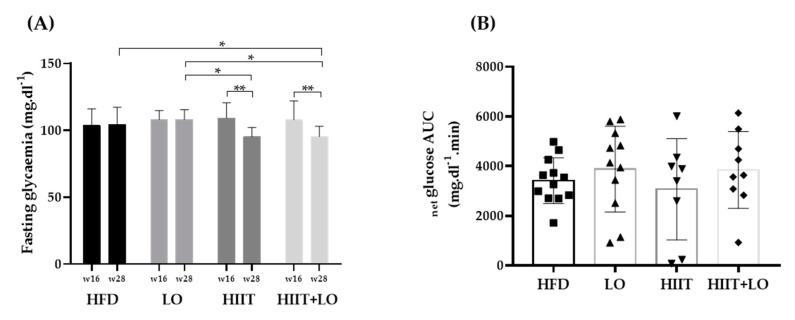
Fasting glycemia and net glucose AUC during the oral glucose tolerance test at week 28 (end of phase 2) in the four groups: HFD (*n* = 12), LO (*n* = 11), HIIT (*n* = 8), and HIIT+LO (*n* = 9). (**A**) Fasting glycemia (mg.dL^−1^) at week 16 and week 28 in the four groups. (**B**) Net glucose AUC (mg.dL^−1^.min^−1^) at week 28. AUC: Area under the curve. * *p* < 0.05, ** *p* < 0.01.

**Figure 6 nutrients-13-00788-f006:**
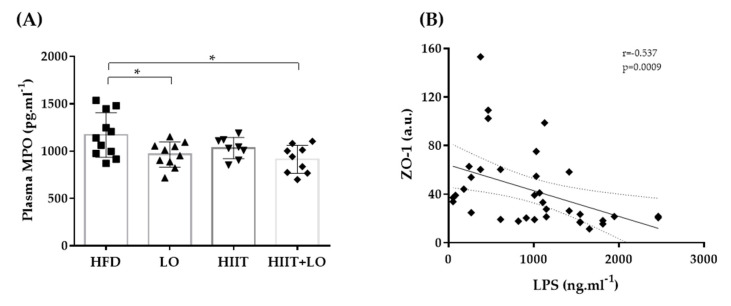
(**A**) Plasma myeloperoxidase (MPO) levels in the four groups and (**B**) correlation of lipopolysaccharide (LPS) and ZO-1: HFD (*n* = 12), LO (*n* = 11), HIIT (*n* = 8), and HIIT + LO (*n* = 9) at the end of phase 2 (training and/or LO supplementation for 12 weeks). * *p* < 0.05.

**Figure 7 nutrients-13-00788-f007:**
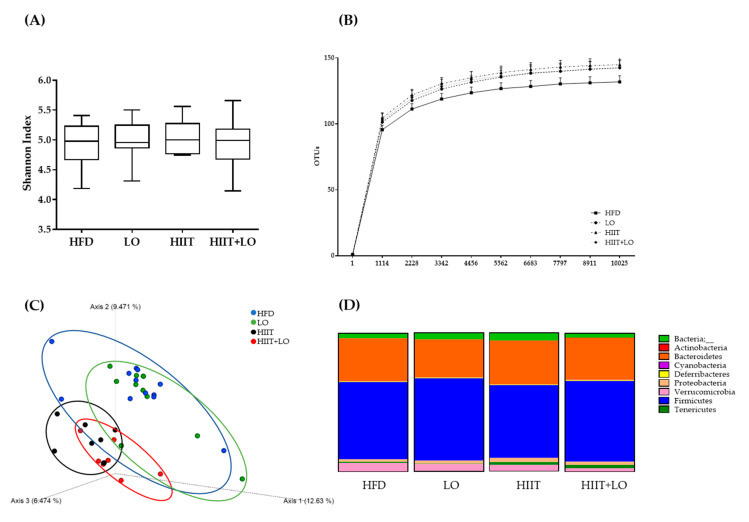
Mucosa-associated microbiota composition analyzed by 16S rRNA gene sequencing in colon DNA samples (Illumina MiSeq system) from the four groups: HFD (*n* = 12), LO (*n* = 11), HIIT (*n* = 7), and HIIT + LO (*n* = 8) at the end of phase 2 (training and/or LO supplementation for 12 weeks). (**A**) Shannon index, (**B**) operational taxonomic units (OTUs) with a rarefaction depth of 10,025 sequences, (**C**) unweighted UniFrac analysis, and (**D**) phylum distribution (%).

**Figure 8 nutrients-13-00788-f008:**
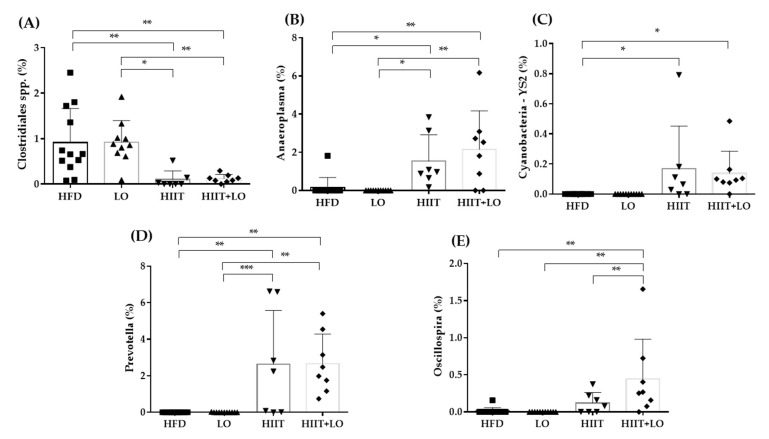
Relative abundance (%) of specific bacterial types in the mucosa-associated microbiota of the four groups: *Clostridiales* spp (**A**), *Anaeroplasma* (**B**), Cyanobacteria YS2 (**C**), *Prevotella* (**D**), *Oscillospira*. (**E**). HFD (*n* = 12), LO (*n* = 11), HIIT (*n* = 7), and HIIT + LO (*n* = 8) at the end of phase 2 (training and/or LO supplementation for 12 weeks). * *p* < 0.05, ** *p* < 0.01, *** *p* < 0.001.

**Figure 9 nutrients-13-00788-f009:**
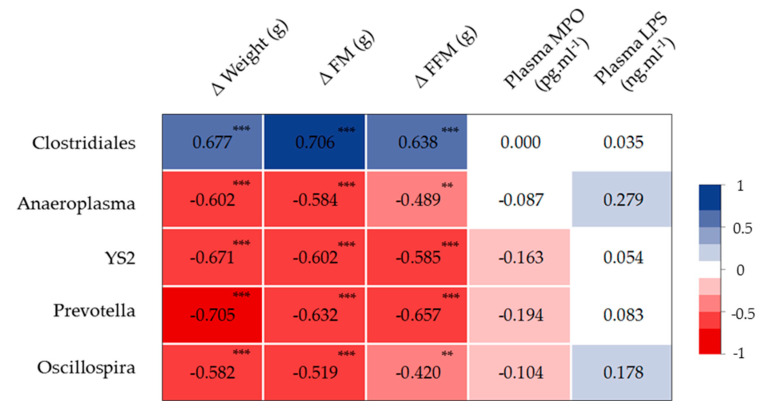
Heat map showing the association between the abundance of the indicated microbiota components, body composition, plasma myeloperoxidase (MPO), and LPS concentration. ** *p* < 0.01, *** *p* < 0.001. ΔFM and ΔFFM, changes in fat mass and fat-free mass between week 28 and week 16, respectively. Color scale indicates positive (blue) to negative (red) correlation.

**Table 1 nutrients-13-00788-t001:** Fatty acid content of the two diets (%). LFD: low-fat diet, HFD: high-fat diet, LO: linseed oil (for the LO groups: HFD + LO ± HIIT), SFA: saturated fatty acids, MUFAs: monounsaturated fatty acids, PUFAs: polyunsaturated fatty acids, LA: linoleic acid (C18:2 n-6), AA: arachidonic acid (C22:4 n-6), ALA: α-linolenic acid (C18:3 n-3), EPA: eicosapentaenoic acid (C20:5 n-3), DHA: docosahexaenoic acid (C22:6 n-3).

	LFD	HFD	LO
Total SFAs	20.67	35.24	34.95
Total MUFAs	34.01	43.54	42.99
Total PUFAs	45.22	20.69	21.85
Total n-6 PUFAs	42.90	20.10	16.35
LA	42.90	19.63	15.95
AA	0.00	0.07	0.07
Total n-3 PUFAs	2.32	0.65	5.50
ALA	2.32	0.65	5.45
EPA	0.00	0.00	0.00
DHA	0.00	0.00	0.00
Ratio n-6/n-3 PUFAs	15.49	29.43	2.97

**Table 2 nutrients-13-00788-t002:** Zonula occludens 1 (ZO-1) and occludin expression (a.u.) by western blotting of colon protein extracts in the four groups: HFD (*n* = 12), LO (*n* = 11), HIIT (*n* = 8), and HIIT + LO (*n* = 9).

	**HFD (*n* = 12)**	**LO (*n* = 11)**	**HIIT (*n* = 8)**	**HIIT + LO (*n* = 9)**	***p***
ZO-1	52.4 ± 41.7	42.3 ± 27.7	34.6 ± 16.9	40.7 ± 27.3	*0.684*
Occludin	108.3 ± 40.8	105.7 ± 44.9	125.2 ± 45.1	141.7 ± 35.9	0.258

**Table 3 nutrients-13-00788-t003:** Fecal acetate, butyrate and propionate concentration (µmol.g^−1^ feces) in the four groups: HFD (*n* = 12), LO (*n* = 11), HIIT (*n* = 8), and HIIT + LO (*n* = 9) at the end of phase 2 (training and/or LO supplementation for 12 weeks). Total short-chain fatty acids (SCFAs) = acetate + butyrate + propionate.

	HFD (*n* = 12)	LO (*n* = 11)	HIIT (*n* = 8)	HIIT + LO (*n* = 9)	*p*
Acetate	24.3 ± 9.0	24.1 ± 9.6	18.3 ± 4.0	25.1 ± 9.0	*0.191*
Butyrate	4.5 ± 2.9	4.1 ± 1.1	2.4 ± 1.1	4.2 ± 3.0	*0.237*
Propionate	2.1 ± 1.1	1.6 ± 0.8	0.9 ± 0.5	1.3 ± 0.6	*0.051*
Total SCFAs	30.8 ± 11.1	29.8 ± 9.9	21.5 ± 4.6	30.1 ± 10.4	*0.144*

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
