# Peer review of "High-Intensity Interval Training and α-Linolenic Acid Supplementation Improve DHA Conversion and Increase the Abundance of Gut Mucosa-Associated Oscillospira Bacteria"

_nutrients, 2021, doi:10.3390/nu13030788_

Round 1
Reviewer 1 Report
The manuscript “High-intensity interval training and α-linolenic acid supplementation improve DHA conversion and 3 increase the abundance of gut mucosa-associated beneficial Oscillospira bacteria” by Claire et al. describes an analysis of the effects of linseed oil and/or exercise on the microbiota and host adiposity, glucose metabolism and inflammatory parameters in an experimental diet-induced obesity rat model. The experimental setup is very straight forward and the analyses were conducted thoroughly. I did not find any main criticisms and congratulate the authors on this interesting work. However, I do suggest few minor changes so that the conclusions better match the data.
1.) The title and some occasions in the text suggest positive or bad attributes connected to certain bacteria, e.g. beneficial Oscillospira. To my knowledge, the role of Oscillospira is not clear today and thus I suggest to remove these attributes such as “beneficial”, but rather state the associations as they are.
2.) The title and main text argues heavily on the role of HIIT. However, the used exercise protocol does not really match classical HIIT with very short and high intense intervals, e.g. 10-60 seconds. I would therefore refrain from terming this as HIIT but rather interval training.
3.) In relation to the previous comment, if the authors intend to make an argumentation for an effect of HIIT or interval training and their better performance compared to classical endurance training, then they need to include this comparison in their experimental setup by adding endurance as well as endurance + LO groups. Instead of adding these groups I would simply rephrase the effects as “exercise” instead of “HIIT” (currently) or “interval training” (see above).
In the bioinformatic description of the amplicon sequencing data, the rarefaction depth is missing. This should be added.
Overall, this study is interesting, well written and only requires minor textual changes. I therefore recommend a minor revision.
Author Response
Reviewer #1: The manuscript “High-intensity interval training and α-linolenic acid supplementation improve DHA conversion and 3 increase the abundance of gut mucosa-associated beneficial Oscillospira bacteria” by Claire et al. describes an analysis of the effects of linseed oil and/or exercise on the microbiota and host adiposity, glucose metabolism and inflammatory parameters in an experimental diet-induced obesity rat model. The experimental setup is very straight forward and the analyses were conducted thoroughly. I did not find any main criticisms and congratulate the authors on this interesting work. However, I do suggest few minor changes so that the conclusions better match the data.
We thank the reviewer for his/her comment.
- The title and some occasions in the text suggest positive or bad attributes connected to certain bacteria, e.g. beneficial Oscillospira. To my knowledge, the role of Oscillospira is not clear today and thus I suggest to remove these attributes such as “beneficial”, but rather state the associations as they are.
We agree. The title has been changed according to the comment and the word “beneficial” has also been removed from the abstract and the text.
- The title and main text argues heavily on the role of HIIT. However, the used exercise protocol does not really match classical HIIT with very short and high intense intervals, e.g. 10-60 seconds. I would therefore refrain from terming this as HIIT but rather interval training.
High-intensity interval training (HIIT) involves alternating short bursts of high intensity exercise with recovery periods or light exercise. In 2014, Weston et al. defined HIIT as intervals of exercise at 85–95% of the peak heart rate and recovery at 70% of the peak hearth rate, without any specification concerning the duration of each interval. In fact the duration of intervals can differ greatly in various HIIT protocols (8s to 5 min, Maillard et al. Sports Med, 2018; Weston et al. 2014). Our laboratory has been focusing on the effects of HIIT on body composition for at least 6 years (Human studies: Maillard et al. 2016, Maillard et al. 2018 a and b; Dupuit et al. 2020 a and b; Animal studies: Groussard et al. 2019, Maillard et al. 2019, Chavanelle et al. 2017). The HIIT protocols used in Maillard et al. 2019 and Groussard et al. 2019 were performed with Zucker rats and were exactly the same as the protocol of this study (6 x [3 min at 10m.min-1 x 4 min at 18m.min-1]). Thus, in continuity with our previous studies on the effect of HIIT on body composition and to compare our present results with our previous publications, we are somewhat reluctant to change HIIT to interval training. We hope that you will understand our choice.
- In relation to the previous comment, if the authors intend to make an argumentation for an effect of HIIT or interval training and their better performance compared to classical endurance training, then they need to include this comparison in their experimental setup by adding endurance as well as endurance + LO groups. Instead of adding these groups I would simply rephrase the effects as “exercise” instead of “HIIT” (currently) or “interval training” (see above).
As previously explained, the HIIT modality is an important topic of our laboratory. In a former study performed in Zucker rats (Maillard et al. PLOS One, 2019), we compared moderate intensity continuous training (MICT) and HIIT and found greater body composition improvement after HIIT training. I quote “In obese Zucker rats, HIIT and MICT improved inflammation and glucose metabolism but only HIIT decreased total and visceral fat mass”. The aim of the present study was to establish whether HIIT and/or α-linolenic acid supplementation might prevent obesity disorders, particularly by modulating the mucosa-associated microbiota. Thus, it appeared important to us i) to justify the choice of exercise modality in the Discussion section by comparing MICT and HIIT and ii) to conclude that HIIT (and not generally “exercise”) combined with α-linolenic acid supplementation induced such effects.
In the bioinformatic description of the amplicon sequencing data, the rarefaction depth is missing. This should be added.
The rarefaction depth has been added in the legend ( Figure 7B; P14L495).
Overall, this study is interesting, well written and only requires minor textual changes. I therefore recommend a minor revision.

Reviewer 2 Report
Comments to the Author:
The present manuscript by Plissonneau et al., analyzed “High-intensity interval training and α-linolenic acid supplementation improve DHA conversion and increase the abundance of gut mucosa-associated beneficial Oscillospira bacteria”. Overall, the authors attempted to evaluate the effects of a 12-week intervention program that included physical activity, n-3 PUFAs supplementation through addition of linseed oil in the diet, or both on body composition and metabolic profile changes in a rodent model of obesity. The manuscript is interesting, covers a topic that has not been well studied. In additions, the manuscript is well written and reads easily. The experiments appear to be sound, well planned and carried out. Nevertheless, and in spite of the significant amount of work performed, some weaknesses need to be addressed:
Major comments:
- The authors have to justify with bibliographic references the recommended doses, as well as the 12 weeks duration of the study and why they used the HIIT training programme.
- The authors should add more information about the slaughter process used in rats. In the same way, they should include more information about how the rats were kept, the type of used cages, the number of animals per cage, etc.
- Did the authors change the circadian cycle of the rats? When the authors carry out the experiments, during the day or at night?
- What explanation could the authors provide to the sudden death of the rats?
- Was a familiarization period there to provide the adaptation of the rats with the treadmill? If there was not a familiarization period, this could explain why some rats did not want to run in the treadmill.
- They were individually housed??? The authors should take into account for future studies that these kinds of cages are stressing for the animals and they must socialize with others animals from the same species.
- How was controlled after 16 weeks that the rats were overweight?
Minor comments:
P2 L68: Add reference.
P2 L72-73: These two sentences are not related.
P8 L368: The 12 mice
P16 L553: Add reference.
P16 L556: Add reference.
P16 L 564: Add reference.
P17 L 624: Add reference.
P18 L 640: Add reference
P18 L648-649: Could the authors justify this with a published study?

Author Response
Reviewer #2 : The present manuscript by Plissonneau et al., analyzed “High-intensity interval training and α-linolenic acid supplementation improve DHA conversion and increase the abundance of gut mucosa-associated beneficial Oscillospira bacteria”. Overall, the authors attempted to evaluate the effects of a 12-week intervention program that included physical activity, n-3 PUFAs supplementation through addition of linseed oil in the diet, or both on body composition and metabolic profile changes in a rodent model of obesity. The manuscript is interesting, covers a topic that has not been well studied. In additions, the manuscript is well written and reads easily. The experiments appear to be sound, well planned and carried out. Nevertheless, and in spite of the significant amount of work performed, some weaknesses need to be addressed:
We thank the reviewer for his/her comments.
Major comments:
The authors have to justify with bibliographic references the recommended doses, as well as the 12 weeks duration of the study and why they used the HIIT training programme.
Concerning the dose used in our study, the objective was to incorporate α-linolenic acid supplementation to equilibrate the n-6/n-3 PUFA ratio. As mentioned in the Discussion section of the manuscript, we chose a physiological dose and not a supra-physiological dose which is often preferred in the literature. As scientific support, we used the French lipid recommendations from the Agence Nationale de Sécurité Sanitaire de l’alimentation, de l’environnement et du travail (ANSES, 2011). This report recommends a n-6/n-3 PUFA ratio <4 as proposed by Simopoulos and collaborators in 2002. This reference has been added in the Discussion to justify our diets.
The HIIT modality is an important topic of our laboratory. In a previous study performed with Zucker rats (Maillard et al. PLOS One, 2019), we compared moderate intensity continuous training (MICT) and HIIT and found greater body composition improvement after HIIT training. We quote “In obese Zucker rats, HIIT and MICT improved inflammation and glucose metabolism but only HIIT decreased total and visceral fat mass”. The aim of the present study was to determine whether HIIT and/or α-linolenic acid supplementation might prevent obesity disorders, particularly by modulating the mucosa-associated microbiota. We modified the paragraph introducing HIIT in the Discussion section to justify the use of this modality.
In the study by Maillard et al. (2019), Zucker rats performed the same HIIT modality (i.e. 4 days per week, each session including 6 x [3 min at 10m.min-1 x 4 min at 18m.min-1]), but for 10 weeks. With this protocol duration, we found significant body composition changes. In the present study, we made the choice to add two additional weeks of exercise for two reasons:
1. First, De Araujo et al. (2016) defined a short-term HIIT program when its duration was <6 weeks and a long-term HIIT programs when its duration was ≥12 weeks, and now 12-week protocols are more frequent (Khalafi et al. Nutrients, 2020)
2. Nutritional supplementation-induced gut microbiota modulation needs at least 2 weeks of treatment and generally lasts less than 24 weeks (Caesar et la., 2015, Ghosh et al., 2013, Millman et al., 2019, Patterson et al., 2014, Ramos-Romero et al., 2017, Yu et al., 2014). The duration of 12 weeks was a good compromise to detect significant gut microbiota alterations.
The authors should add more information about the slaughter process used in rats. In the same way, they should include more information about how the rats were kept, the type of used cages, the number of animals per cage, etc.
Some information has been added in the manuscript.
P3 L122. The paragraph 2.1 has been completed as follows:
Sixty 8-week-old male Wistar rats (Charles River Laboratory, France) were used for this study. They were individually housed in a temperature-controlled (22 ± 2 °C) room with reversed 12h light/dark cycle and free access to food and water. All experiments were approved by the local ethics committee (C2EA-02, Auvergne, France; APAFIS 16090-2018071208306750), and were performed in accordance with the European animal welfare regulations and guidelines (European Directive 2010/63/EU on the protection of vertebrate animals used for experimental and scientific purposes). All efforts were made to protect animal welfare and to minimize suffering at each step of the protocol. Animals were sacrificed by cervical dislocation after isoflurane anesthesia.
Did the authors change the circadian cycle of the rats? When the authors carry out the experiments, during the day or at night?
Rats were housed in a room with reversed 12h light/dark cycle (P3 L125) (light on at 7PM until 7AM). Thus, the trainings were carried out during the night period for the animals.
What explanation could the authors provide to the sudden death of the rats?
The death of the rats during the study was caused by wrong gavage manipulation (n=3) during the oral glucose tolerance test.
Was a familiarization period there to provide the adaptation of the rats with the treadmill? If there was not a familiarization period, this could explain why some rats did not want to run in the treadmill.
Animals assigned to the HIIT groups were familiarized on the treadmill with a progressive increase of speed and running time for 2 weeks. After these 2 weeks, we decided to stop the familiarization period, and kept only “the runners”.
Some information has been added in the manuscript.
P15 L172. In the paragraph 2.4 “Training design”:
Rats in the HIIT (n=12) and HIIT+LO (n=12) groups underwent a 2-week acclimation on a treadmill (Matsport, France) with a progressive increase of speed and running time. “No runner” rats were excluded at the end of this period. Then, rats performed a 12-week HIIT program, 4 days per week, as follows: 6 x [3 min at 10m.min-1 x 4 min at 18m.min-1], for a total of 42 min/training (Figure 1).
They were individually housed??? The authors should take into account for future studies that these kinds of cages are stressing for the animals and they must socialize with others animals from the same species.
We agree that keeping rats individually induces stress, but all groups were in the same conditions. Furthermore, we needed to measure food intake during the protocol to establish potential differences of energy intake among groups. To limit stress, cages were near to each other and the rats could smell and ear the other rats. In each cage, the amount of play area was increased. This procedure was discussed and approved by the local ethics committee (C2EA-02, Auvergne, France; APAFIS 16090-2018071208306750), and was performed in accordance with the animal welfare regulations and guidelines (European Directive 2010/63/EU on the protection of vertebrate animals used for experimental and scientific purposes).
How was controlled after 16 weeks that the rats were overweight?
Weight was monitored during the 16 first weeks and became significantly higher in the HFD group from week 14. Fat mass was measured by EchoMRI at week 1 and week 16, and reached higher values (g and %) in the HFD group at the end of this period. On the basis of literature data and our experience, we thought that 16 weeks would have been enough to reach the obesity state. Unfortunately, we only obtained a pre-obesity state (an excess of weight and FM) associated with metabolic disorders (as shown by the higher “Glucose AUC” in HFD vs LFD during the OGTT).
Minor comments:
P2 L68: Add reference.
For review: Wewege et al., 2018
P2 L72-73: These two sentences are not related.
P2 L64 to 78. We changed the two sentences that were not related in the manuscript as below:
Endurance training is an effective strategy to prevent overweight and obesity [5]. This exercise modality includes low- to moderate-intensity continuous training, and high intensity interval training (HIIT). Moderate-intensity continuous training remains the most recommended physical activity [6], but HIIT is now also suggested for people with obesity [7]. HIIT (i.e. repeated bouts of high-intensity effort at >80-85% of the peak heart rate followed by varied recovery times [8]) is a time-efficient and safe strategy to reduce total fat mass (FM), particularly subcutaneous and intra-abdominal FM [9,10]. HIIT programs also decrease inflammation and improve insulin sensitivity [11–13]. In addition, several supplements and nutritional interventions may enhance HIIT effects by increasing energy metabolism, or by modulating the adaptive response during recovery [14]. However, no data is available concerning the potential effect of n-3 polyunsaturated fatty acid supplementation on HIIT adaptations.
P8 L368: The 12 mice
We corrected the sentence.
P16 L553: Add reference.
Esser et al., 2014 and Iacobini et al., 2019
P16 L556: Add reference.
San-Cristobal et al., 2020, Ling and Rönn 2019, Rohde et al., 2019, Nicolaidis et al., 2019
P16 L 564: Add reference.
San-Cristobal et al., 2020, Nicolaidis et al., 2019
P17 L 624: Add reference.
Nicholls and Hazen, 2005
P18 L 640: Add reference
P18L640-641. In our study, HIIT+LO supplementation modulated β-diversity of the mucosa-associated microbiota.
P18 L648-649: Could the authors justify this with a published study?
Most studies used the most bioactive forms of n-3 PUFAs, i.e. EPA and DHA (Cf. review: Costantini et al., 2017). These fatty acids can be synthesized from the dietary precursor and essential fatty acid, α-linolenic acid (ALA). However, as the synthesis pathway requires several elongation and desaturation chemical reactions, the conversion of the two active forms in mammals is less efficient than dietary uptake. However, foods rich in these fatty acids are not widespread, therefore, EPA and DHA are widely used as nutritional supplements, often as nutraceuticals.
The study by Costantini et al. (2017) is mentioned in the Discussion section (line 653).

Round 2
Reviewer 2 Report
For the author:
I appreciate authors’ effort. The authors have obviously spent considerable time revising the manuscript and their hard work is clearly paying off. This manuscript is drastically improved from the original submission. The message is very clear, the language is much more clean, and the issues in the first version were corrected. Besides, the authors have answered all my comments successfully. For this reason, I encourage to editor to consider this manuscript for publication for the interesting value of the study realized, that now it is a much more robust study.
